# Establishment of a Direct PCR Assay for Simultaneous Differential Diagnosis of African Swine Fever and Classical Swine Fever Using Crude Tissue Samples

**DOI:** 10.3390/v14030498

**Published:** 2022-02-28

**Authors:** Tatsuya Nishi, Kota Okadera, Katsuhiko Fukai, Miwa Yoshizaki, Ai Nakasuji, Syuji Yoneyama, Takehiro Kokuho

**Affiliations:** 1Exotic Disease Research Station, National Institute of Animal Health, National Agriculture and Food Research Organization, 6-20-1 Josui-honcho, Kodaira 187-0022, Tokyo, Japan; ultra1124@affrc.go.jp (T.N.); sry46-s5y492602@city.saitama.lg.jp (K.O.); fukai@affrc.go.jp (K.F.); 2Takara Bio Inc., Nojihigashi 7-4-38, Kusatsu 525-0058, Shiga, Japan; yoshizakim@takara-bio.co.jp (M.Y.); nakasujia@takara-bio.co.jp (A.N.); 3Tochigi Prefectural Central District Animal Hygiene Service Centre, Hiraideko-gyo-danchi 6-8, Utsunomiya 321-0905, Tchigi, Japan; yoneyamas01@pref.tochigi.lg.jp

**Keywords:** African swine fever, classical swine fever, real-time reverse-transcription polymerase chain reaction, multiplex detection, crude nucleic acid extraction

## Abstract

African swine fever (ASF) and classical swine fever (CSF) are contagious swine diseases that are clinically indistinguishable from each other; hence, reliable test methods for accurate diagnosis and differentiation are highly demanded. By employing a buffer system suitable for crude extraction of nucleic acids together with an impurity-tolerant enzyme, we established a multiplex assay of real-time reverse-transcription polymerase chain reaction (rRT-PCR) for simultaneous detection of ASF virus (ASFV), CSF virus (CSFV) and swine internal control derived genes in a sample without the need for prior purification of viral nucleic acids. We applied this method to test serum and tissue samples of infected pigs and wild boars and compared the statistical sensitivities and specificities with those of standard molecular diagnostic methods. When a serum was used as a test material, the newly established assay showed 94.4% sensitivity for both and 97.9 and 91.9% specificity for ASFV and CSFV detection, respectively. In contrast, the results were 100% identical with those obtained by the standard methods when a crude tissue homogenate was used as a test material. The present data indicate that this new assay offers a practical, quick, and reliable technique for differential diagnosis of ASF and CSF where geographical occurrences are increasingly overlapping.

## 1. Introduction

African swine fever (ASF) and classical swine fever (CSF) are contagious viral diseases that affect domestic and wild suids. Both diseases cause devastating damage to livestock industries due to high morbidity and mortality rates, and potential for severe economic loss in affected countries, along with a huge negative impact on international trade of pigs and pork products [1,2]. Currently, the epidemic of ASF is progressing worldwide, including in China, the world’s largest pig producing country [3,4]. CSF is also spreading widely in Japan and other Asian countries [2]. In 2018, CSF re-emerged in Japan for the first time in 26 years and raised public concern nationwide [5]. Domestic pigs and other susceptible suid species, especially wild boar, can be the source for virus intrusion into pig farms during the ASF and CSF pandemic [6,7]. These two diseases are indistinguishable from each other by visual inspection of affected animals [1,2]. In ASF and CSF, affected pigs similarly show pyrexia, inappetence, dullness, haemorrhage, and cyanosis. Moreover, both diseases abrogate the immune systems of host animals and lead to concurrent infections with other viral and bacterial pathogens, which may complicate clinical manifestations. Therefore, a tentative diagnosis by clinical observation or post-mortem inspection must be confirmed by laboratory investigation.

ASF virus (ASFV) is a large, enveloped double-stranded DNA virus classified into the genus *Asfivirus* of the family Asfarviridae [8]. ASFV can be detected by virus isolation using porcine leukocytes or bone marrow cell cultures, visualisation of viral antigens in a tissue smear, cryosection by direct fluorescent antibody test (FAT), or detecting virus-derived genes by polymerase chain reaction (PCR) as described in the World Organization for Animal Health (OIE) Terrestrial Manual [9]. Among these techniques, PCR is currently the most sensitive technique for diagnosing ASF, especially at the early stages of infection [10,11,12]. In addition, PCR is particularly useful if a submitted specimen is not applicable to virus isolation or FAT due to contamination or putrefaction. In an outbreak of a less virulent strain of ASFV, the status of viraemia continues for several weeks in affected pigs; thus, PCR can also be very useful for detecting recovering animals at a later stage of infection [13]. Furthermore, simultaneous detection of the internal control (IC) gene enables the exclusion of false negatives as an index of the success or failure of the PCR test process [14].

CSF virus (CSFV) is a single positive-stranded RNA virus belonging to the genus *Pestivirus*, and the family Flaviviridae [15]. The genus *Pestivirus* includes bovine viral diarrhoea virus (BVDV) and border disease virus (BDV), which mainly affect cattle and sheep, respectively, but can also naturally infect pigs [16]. The untranslated region (UTR) of the CSFV genome is highly conserved among the viruses within the genus [17]; hence, this region can be a universal target gene for *Pestivirus* detection. Such genetic similarity in the genome sequences within the genus leads to difficulties in employing PCR or other conventional methods, such as FAT and antigen-ELISA, to discriminate CSFV from BVDV and BDV. TaqMan real-time PCR (rPCR) based methods combined with specific primers and an optimised probe for CSFV detection can overcome this problem [18]. These methods, however, are costly and require time- and labour-consuming procedures to prepare nucleic acid fractions from test specimens.

A reliable and practical test method could be key for prompt implementation of counter measures against ASF and CSF pandemics at an early stage of occurrence. In the present study, we established a simple and rapid test method for specific and simultaneous identification of ASFV and CSFV and validated the procedure using clinical samples of experimentally and naturally infected animals in comparison with other authentic PCR assays for the diagnosis of both diseases. Our new assay, which is based on a multiplex TaqMan real-time reverse transcription PCR (rRT-PCR), enables quick differential diagnosis of both distinctively important swine diseases, ASF and CSF, in less than two hours with high accuracy.

## 2. Materials and Methods

### 2.1. Viruses

The virus strains, CSFV/JPN/1/2018 (genotype 2.1) and CSFV/JPN/27/2019 (genotype 2.1), were isolated from clinical samples of infected pigs using porcine kidney (CPK) cell cultures at our laboratory [19]. The ASFV strains, Armenia/07 (genotype II), Kenia/05 (genotype X), and España/75 (genotype I), were obtained from the OIE reference laboratory for ASF (Universidad Complutense de Madrid, Spain). Another ASFV strain, AQS-C-1-22 (genotype II), which was isolated from an illegally imported pork product seized at an international airport by the animal quarantine service (AQS), was propagated in immortalised porcine macrophage (IPKM) cell line [20]. ASFV containing tissue homogenate stocks were prepared as follows: 6-week-old Landrace–Large White–Duroc (LWD) pigs (IDs 1705-1 to 6, 2105-9, 10 as indicated in Table 1) were infected with different strains of ASFV. At 5–7 days post-infection (dpi), infected pigs were euthanised, and tissue samples (listed in Table 1) were collected. Cleared supernatants of 10% homogenate of tissue sample were obtained after centrifugation at 8000× *g* for 10 min, aliquoted and stored at −80 °C until used for animal experiments as inoculum or test specimen. All the experiments were conducted in a biosafety level (BSL) 3 facility of the National Institute of Animal Health (NIAH).

### 2.2. Clinical Samples of Field Cases

Clinical samples of symptomatic pigs and wild boars found dead or captured in the field were submitted to prefectural animal hygiene service centers (AHSCs) for passive surveillance of ASF/CSF during 2018–2021. These samples, which include tonsil, spleen, kidney, blood, and serum, were taken by veterinary officials according to the guidelines of the Domestic Animal Infectious Diseases Control Act, then, subjected to nucleic acid purification before conducting standard conventional RT-PCR for CSFV [17] and PCR for ASFV [21] with slight modification at AHSCs. The serum and tissue samples that appeared to be positive for CSFV were sent to our laboratory for confirmatory diagnosis. Information regarding all the clinical samples used in this study are indicated in Appendix A.

### 2.3. Clinical Samples of Experimentally Infected Animals

Serum samples of CSFV infected animals were collected as described in our previous reports [22,23]. Briefly, two each of domestic pigs (IDs 1810-2, 11) and pig-boar hybrids (IDs 1905-4, 5) were intraorally inoculated with CSFV/JPN/1/2018 and CSFV/JPN/27/2019, respectively, and bled daily since the day of inoculation. To collect samples of ASFV infected animals, five pigs (IDs 1806-3 to 7) were intranasally or intraorally inoculated with Armenia/07. Two pigs (IDs 2105-9, 10) and three wild boars (IDs 2101-1 to 3) were intraorally inoculated with AQS-C-1-22 and Armenia/07, respectively. In another experiment, two wild boars (IDs 2002-1, 2) were intramuscularly inoculated with Armenia/07. Three other healthy wild boars (IDs 2002-3 to 5) were cohabited with the infected ones immediately after inoculation. Serum samples were collected daily from the jugular vein. At post-mortem examinations, tonsil, spleen, kidney, and mesenteric lymph nodes were collected from each animal and homogenized using a Micro Smash MS-100R (TOMY SEIKO, Tokyo, Japan). The homogenates were emulsified in Dulbecco’s modified Eagle’s medium to yield a 10% (wt/vol) tissue homogenate and centrifuged at 8000× *g* for 10 min to collect cleared supernatants. All the animal experiments performed in this study were approved by the animal care and use committee of the NIAH (authorisation numbers: 17-004, 18-047).

### 2.4. Preparation of Test Samples for Multiplex rRT-PCR

Test samples used for the multiplex rRT-PCR assays were prepared as follows: serum samples and supernatants of tissue homogenate obtained from pigs and wild boar were mixed with an equal volume of Lysis Buffer S (LBS; code no. 9811, Takara Bio Inc., Shiga, Japan), which is capable of extracting nucleic acids in serum specimens. This process was completed after incubation for 5 min at room temperature (approximately 25 °C), then, samples were left on ice until use to avoid unfavourable destruction of the sample according to the manufacture’s instructions. In our preliminary experiment, up to 4 μL of LBS in a volume of 25 μL of reaction mixture did not affect the amplification of ASFV, CSFV, and IC target sequences in the CSFV/ASFV positive control DNA (Takara Bio Inc.).

### 2.5. Multiplex rRT-PCR Amplification

The reaction mixture was prepared in a volume of 25 µL containing 2 µL of LBS-treated crude test sample, 2× concentrated rRT-PCR mix containing reverse transcription (RT), PCR, and uracil N-glycosylation (UNG) enzymes, and a pre-mixed primer/probe (CSFV/ASFV Direct RT-qPCR Mix & Primer/Probe with ROX Reference Dye; code no. RC212A), which were produced and supplied by Takara Bio Inc. The primer/probe sets for ASFV, CSFV, and IC detection were composed of oligonucleotides and target gene-specific TaqMan probes modified with FAM, Cyanine5, and HEX, respectively, and dark quencher.

Instead of crude test samples, CSFV/ASFV positive control DNA (code no RC215A; Takara Bio Inc.), which contains artificial DNA templates designed based on the deposited genomic sequences of ASFV_HU_2018 and CSFV HY78 strains (DDBJ accession nos. MN715134 and MH979231, respectively), was used for the validation of the assay. The multiplex rRT-PCR assays were performed using an Applied Biosystems 7500 Fast Real-Time PCR System and QuantStudio 5 Real-Time PCR System (Life Technologies, Carlsbad, CA, USA) as follows: 25 °C for 10 min, 52 °C for 5 min, one cycle; 95 °C for 10 s, one cycle; 95 °C for 5 s, 56 °C for 30 s, 45 cycles with fluorescence reading in the FAM, Cyanine5, and VIC channels at the end of each cycle.

### 2.6. Conventional Real-Time PCR (rPCR) and RT-PCR

The sensitivity and specificity for each viral template detected by multiplex rRT-PCR were compared with those by either standard singleplex rPCR for ASFV or conventional RT-PCR for *Pestivirus* with purified nucleic acids as a reference. Nucleic acids were purified from serum or tissue homogenate samples of pigs and wild boar with the High Pure Viral Nucleic Acid Kit (Roche Diagnostics, Basel, Switzerland). Note that for the comparison of the sensitivities for ASFV and CSFV assays, 50 μL of the sample (initial input) was used for nucleic acid purification, whereas 100 μL of the sample was used for purification in other experiments. Purified nucleic acid fractions were eluted in 50 μL of elution buffer according to the manufacturer’s instructions and stored at −20 °C until further use.

For the singleplex rPCR for ASFV, reactions were performed using a TaqMan Fast Advanced Master Mix (Applied Biosystems, Waltham, MA, USA) employing specific primers and a probe modified with FAM in a volume of 20 µL of reaction mixture containing 2 µL of a purified nucleic acid test sample as described elsewhere [21]. The amplification conditions were 50 °C for 2 min, 95 °C for 20 s, followed by 45 cycles of 3 s at 95 °C and 30 s at 58 °C with fluorescence reading in the FAM channel at the end of each cycle.

For *Pestivirus* detection by conventional RT-PCR, 5 µL of a purified viral nucleic acid sample was subjected to RT-PCR using a Superscript III One-Step RT-PCR System with Platinum Taq Polymerase (Life Technologies) and a set of 324 (forward) and 326 (reverse) primers in a total volume of 50 µL of reaction mixture as described by Vilcek et al. [17]. PCR amplification was performed using an ABI GeneAmp PCR System 9700 (Life Technologies) as follows: 55 °C for 30 min, one cycle; 94 °C for 2 min, one cycle; 94 °C for 15 s, 55 °C for 30 s, and 68 °C for 45 s, 35 cycles; and 68 °C for 5 min, one cycle. Amplified products were then separated by electrophoresis on a 1% agarose gel. The products were then stained with ethidium bromide and visualised using UV light transillumination. To confirm specific amplification, restriction enzyme digestion with *Bgl*I was performed by AHSCs as described previously [17].

## 3. Results

### 3.1. Analytical Sensitivity and Linearity of the Optimised Multiplex Real-Time RT-PCR Assay

The analytical sensitivity and linearity of the optimised multiplex rRT-PCR assay was determined using serial dilutions of the positive control DNA templates of ASFV, CSFV, and IC as test material. Standard curves (Ct values versus log_10_ DNA copies) demonstrated significant linearity in ASFV, CSFV, and IC target sequence amplification between 10 to 10^6^ copies of each artificial template (correlation coefficient, R^2^ = 0.998, 0.991, and 0.996 for ASFV, CSFV, and IC, respectively; efficiency of amplification, E = 105.293%, 110.625%, and 104.328% for ASFV, CSFV, and IC, respectively, calculated using 7500 Software v2.3 (Life Technologies)) (Figure 1).

### 3.2. Analytical Sensitivities of ASFV and CSFV from Crude Nucleic Acid and Purified Nucleic Acid

Crude and purified nucleic acid test samples prepared from serum and tissue specimens of pigs and wild boar obtained in either experimental or field studies were analysed simultaneously to evaluate statistical accuracies of ASF and CSF diagnoses. When amplification plots of crude and purified test samples, which contained theoretically identical amounts of nucleic acids (equivalent to 1 μL of serum), were analysed by a paired t-test, no significant difference was observed in Ct values for ASF, CSF, and IC between the same serum specimens that were processed differently (Figure 2). This result suggested that crude preparation of nucleic acids is sufficient for this newly established assay as a test sample and that it may not cause any decrease in the sensitivities for ASFV, CSFV, and IC detection compared to the standard test methods.

The outlined procedures of the established system consisting of crude nucleic acid preparation and multiplex rRT-PCR amplification are shown in Figure 3. Representative amplification plots obtained by the multiplex rRT-PCR are shown in Figure 4. Obvious exponential curves of fluorescent signals of FAM and VIC indicative of the amplification of ASFV and IC, respectively, were obtained with crude test samples prepared from serum and spleen homogenates of ASFV-infected wild boar. In the same assay, similar curves of Cyanine5 and VIC signals were observed with crude test samples of sera and tonsil homogenates of dead wild boar suspected to be infected with CSFV by passive surveillance at AHSCs.

### 3.3. Detection of ASFV in the Specimens of Experimentally Infected Pigs and Wild Boar

To evaluate statistical sensitivities for ASFV detection in serum specimens, blood samples were collected daily from two pigs and five wild boars inoculated with ASFV Armenia/07 over the course of an experimental infection. Two microlitres of crude nucleic acid preparation, which contained 1 μL of serum and 1 μL of LBS, was analysed by the newly established multiplex rRT-PCR. On the other hand, 2 μL of purified nucleic acid preparation was also tested by the singleplex ASFV rPCR assay. It is noteworthy that there was a 4-fold difference in the input amount between crude and purified nucleic acid preparations in proportion to the volume of the serum used for the purification. The multiplex rRT-PCR detected ASFV-specific fluorescent signals in the test samples of an infected pig (ID 1806-3) one day earlier than the singleplex ASFV rPCR, whereas the same multiplex assay detected positive signals in the samples of another infected pig (ID 1806-7) one day later than the singleplex assay (Figure 5). The Ct value for ASF in the serum of pig ID 1806-7 at 2 dpi (which was ASF negative in the multiplex rRT-PCR) in the singleplex ASFV rPCR was 39.0 indicating that only a few copies of viral genes were present. For the five experimentally infected wild boars (IDs 2002-1 to 5), both assays started to detect ASFV-positive signals on the same day of infection (Table 1). The present results indicated that the newly established multiplex rRT-PCR was comparable to the authentic test method for ASFV diagnosis when serum specimens were used as a test sample.

Tissue homogenates of tonsil, spleen, kidney, and mesenteric lymph nodes of eight pigs and three wild boars experimentally infected with four different ASFV isolates were also evaluated (Table 1). By using the singleplex ASFV rPCR and multiplex rRT-PCR assays in parallel, identical positive results were observed with the test samples collected from all the ASFV-infected animals regardless of the inoculum. This result demonstrated that the multiplex rRT-PCR presented in this report is compatible with the reference singleplex rPCR assay for ASFV detection.

### 3.4. Detection of CSFV in Experimentally Infected Pigs and Pig-Boar Hybrids

The serum samples of pigs and pig-boar hybrids experimentally infected with Japanese isolates of CSFV were analysed (Table 2). Test results obtained by multiplex rRT-PCR were compared with those obtained by conventional RT-PCR for *Pestivirus* detection [17]. Two microlitres of crude nucleic acid preparation was subjected to multiplex rRT-PCR while 5 μL of purified nucleic acid preparation of the same serum specimens was subjected to the RT-PCR assay at a volume of 25 μL per reaction. It is noteworthy that the input of purified nucleic acids was equivalent to 10 μL of serum, then, 1/5 of amplified products equivalent to 2 μL of serum, was used for visual observation. Hence, there would be a 2-fold difference in input volume of the specimen between the new and conventional assays for CSFV diagnosis. In the multiplex rRT-PCR assay, positive results were detected in the test samples prepared from an infected pig (ID 1806-1) one day earlier than in the conventional RT-PCR assay, while the test sample of another infected pig (ID 1806-2) was detected to be positive on the same day by both assays. The samples of two experimentally infected wild boars (IDs 1905-4, 5) were detected as positive by the new assay 1–2 days earlier than by the conventional assay. This suggested that the sensitivity of the newly established multiplex rRT-PCR assay was higher than that of the standard assay when serum specimens were used as test material.

### 3.5. Validation Study Using Clinical Samples Obtained from Experimental and Field Samples

The detection sensitivity and specificity of the multiplex rRT-PCR were determined using reference samples which were defined as ASFV or CSFV positive or negative by primary diagnosis by the singleplex rPCR (ASFV) and RT-PCR assays (*Pestivirus*). A total of 200 specimens (144 sera and 56 tissue samples) were collected from 114 animals infected with either ASFV or CSFV or healthy animals as listed in Table 1 and Table 2 and Appendix A. Fifty-one out of 53 crude nucleic acid samples of serum specimens (positive; 18, negative; 35) were determined concordantly by rRT-PCR, although one negative sample was determined as positive and the other positive was determined as negative by the same assay. On the other hand, 101 out of 108 test samples of serum were identically determined (positive; 71, negative; 37), whereas three samples that were found negative by the RT-PCR for *Pestivirus* were determined as positive and the other four positive samples were determined as negative and vice versa. It should be noted that the former three samples giving negative results were all from experimentally infected animals with clinical signs and that the latter four positive samples with haemolytic appearance were all from wild boars found dead in the field by passive surveillance.

The statistical sensitivity and specificity were calculated as follows; Sensitivity = TP/(TP + FN), Specificity = TN/(TN + FP). (True positive (TP), a positive result in both methods; False positive (FP), a positive result in the multiplex rRT-PCR, but a negative result in the primary diagnostic method; True negative (TN), a negative result in both methods; False negative (FN), a negative result in the multiplex rRT-PCR, but a positive result in the primary diagnostic method). In summary, the statistical sensitivities of ASFV and CSFV detection using serum as a test sample were 94.4%, while the specificities were 97.1 and 91.9%, respectively (Table 3). All the crude test samples prepared from tissue homogenates, which included 23 ASFV positive, 24 CSF positive, and 12 negative specimens determined by primary diagnostic methods in advance, produced identical results by the multiplex rRT-PCR presented in this report (Table 3).

## 4. Discussion

In this study, we established a rapid, simple test method for simultaneous detection of ASFV and CSFV (Figure 3). First, this method requires a simple procedure for nucleic acid extraction which involves mixing a given tissue specimen with an equal volume of LBS, rather than taking time and effort in DNA/RNA purification. In addition, this method requires only a small amount (usually 2 μL per assay) of serum or tissue emulsion sample as an initial material; hence, the procedure for sample collection can also be greatly simplified. Second, the present method may shorten test procedures for each step of the assay. For example, sample preparation takes 5 min, and the amplification reaction ends within 1 h, this allows quick response to contain a possible outbreak of swine diseases with a huge economic impact. Lastly, we employed techniques to avoid both false-negative and false-positive results, which can be caused by improper operation of test procedures. To exclude false negative results which may occur due to insufficient extraction of nucleic acids or contingent inhibition of RT-PCR reaction, a system for the amplification of an exogenous internal control gene was incorporated. The addition of UNG enzyme can minimise false positive results, which may occur due to the contamination of amplified PCR products. This new test method therefore offers greatly simplified procedures for simultaneous diagnosis of ASF and CSF with considerable reliability.

In this study, we evaluated multiplex rRT-PCR with the use of clinical samples from ASFV- and CSFV-infected and healthy animals. In pigs infected with ASFV and CSFV, the viruses were detected in blood and serum samples at high viral loads in the early stages of infection [10,11,22]. A large amount of virus is also distributed in various organs including tonsil, spleen, and lymph nodes. Hence, sera from symptomatic animals and organ tissues from dead animals are suitable materials for diagnosis and surveillance. The combination of simple extraction procedures and an optimised PCR reagent system for efficient reverse transcription and amplification enables us to detect target genes in a crude test sample without affecting the sensitivity of diagnosis (Table 3). However, in addition to when samples contained few viral genes, false-negative results for CSFV detection were occasionally observed in poorly conditioned samples collected from wild boar found dead in the field (Appendix A). As positive signals for IC were obtained in the amplification plots of these samples, neither extraction of nucleic acids nor RT-PCR reaction appeared to be suppressed. Hence, we deduced that such discrepancies between the new and established methods may be due to the suppression of fluorescence-specific signal detection by haemolysis of the samples. As this new multiplex rRT-PCR is also applicable to purified test samples, we recommend performing nucleic acid purification with an appropriate method prior to the amplification reaction when the quality of the specimen is not suitable for the analysis of crude preparation.

In conclusion, our newly established test method offers a significant enhancement in the diagnosis of two distinct infectious diseases constituting serious threats to suid species. Therefore, this method would play an important role in rapid diagnosis of the diseases and provide a practical approach to conducting large scale field surveillance in endemic countries.

## Figures and Tables

**Figure 1 viruses-14-00498-f001:**
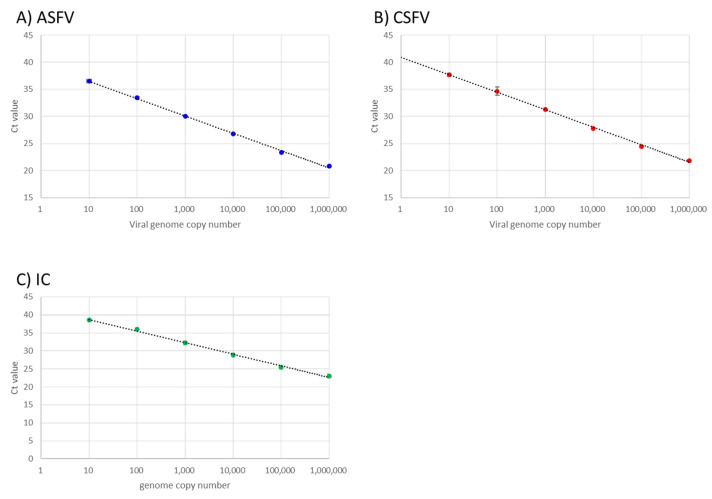
Analytical sensitivity of the developed multiplex real-time reverse transcription (rRT)-PCR for African swine fever virus (ASFV), classical swine fever virus (CSFV), and internal control (IC) detection. Serial dilutions of CSFV/ASFV positive control DNA (Takara Bio Inc.) containing artificial templates of (**A**) ASFV, (**B**) CSFV, and (**C**) porcine GAPDH (IC) were amplified using the newly developed multiplex rRT-PCR system.

**Figure 2 viruses-14-00498-f002:**
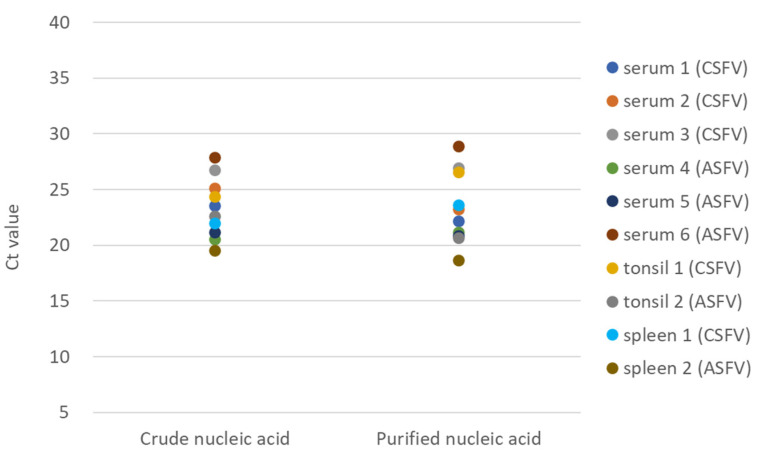
Comparison of the analytical sensitivities of ASFV and CSFV from crude nucleic acid and purified nucleic acid. The sera and tissue homogenate samples obtained from field cases in wild boar and experimentally infected pigs which had been defined as positive using the CSFV/ASFV multiplex rRT-PCR were analysed to evaluate the analytical sensitivities between crude and purified nucleic acid. Sera 1–6 were obtained from wild boar, ID: 1, 3, 4, 2002-2 (4 days post-infection; dpi), 2002-3 (13 dpi), and 2002-5 (9 dpi), respectively, as listed in Table 1 and Table 2. The tonsil and spleen samples were obtained from wild boar, ID: IS-633 and 2101-2, respectively.

**Figure 3 viruses-14-00498-f003:**
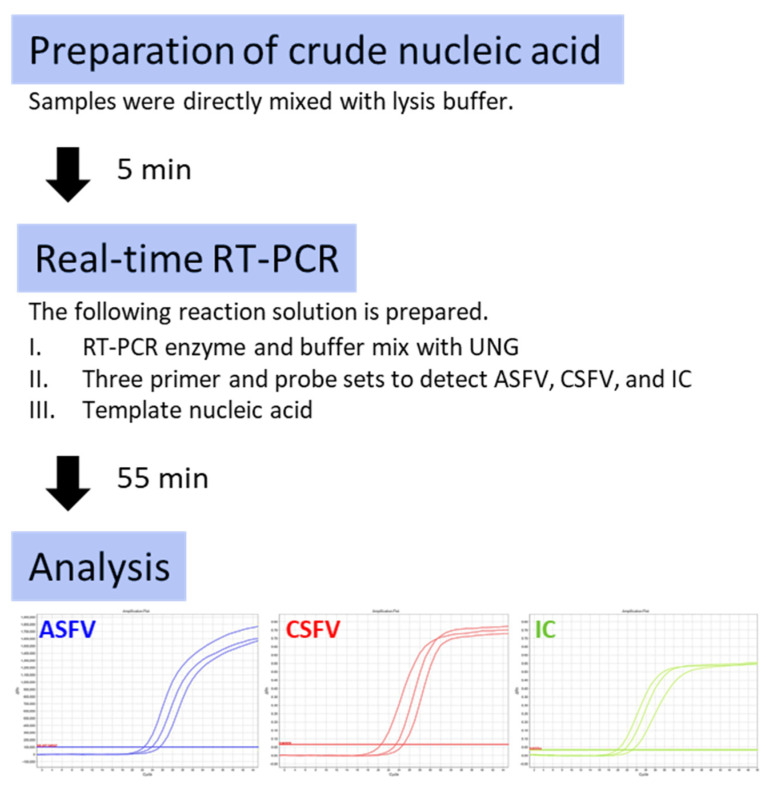
Outline of the direct-multiplex rRT-PCR system for the detection of ASFV and CSFV. The developed direct-multiplex rRT-PCR system requires only three steps to detect ASFV, CSFV, and IC.

**Figure 4 viruses-14-00498-f004:**
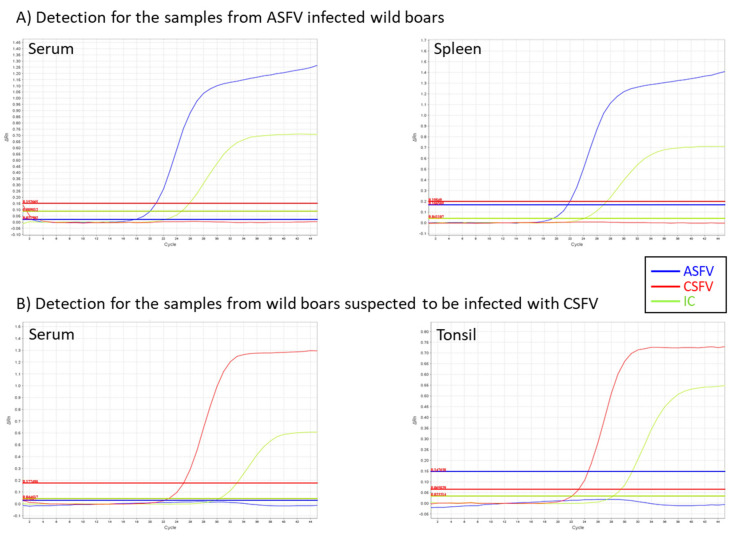
Representative amplification plots of the multiplex rRT-PCR. (**A**) Detection for the serum and tissue homogenate of spleen from ASFV infected wild boar. The serum and spleen were collected from experimentally infected wild boar (ID, 2002–3) and pig (ID: 2105-10) as listed in Table 1. (**B**) Detection of the serum and tissue homogenate of tonsils from wild boar suspected to be infected with CSFV. The serum and tonsil were collected from infected wild boar (ID: YN_174) and (ID: IS-633) in the field as listed in Appendix A.

**Figure 5 viruses-14-00498-f005:**
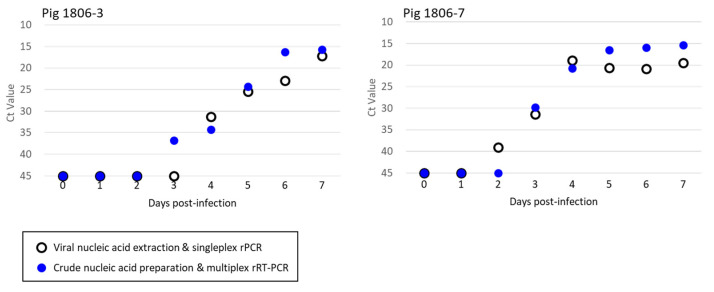
Comparison of the analytical sensitivities between conventional real-time PCR and direct-multiplex rRT-PCR on sera from wild boars experimentally infected with ASFV Armenia/07 strain. Viral nucleic acids were extracted from blood serum samples daily until 7 dpi from two pigs experimentally infected with ASFV Armenia/07 strain and analysed by multiplex RT-PCR (filled symbols) or ASFV simplex rPCR-TaqMan assay (open symbols).

**Table 1 viruses-14-00498-t001:** Detection of ASFV in samples obtained from pigs and wild boars experimentally infected with ASFV isolates.

						Multiplex rRT-PCR *			
		Inoculated	Date of Sampling		Singleplex	CSF	ASF	IC	
Animal	ID	Virus Strain		Sample	rPCR	Cy5	FAM	VIC	Decision
Pig	1806-3	Armenia/07	0 dpi	serum	−	−	−	32.2	negative
			1 dpi	serum	−	−	−	33.7	negative
			2 dpi	serum	−	−	−	34.0	negative
			3 dpi	serum	−	−	36.7	33.3	ASFV
			4 dpi	serum	31.3	−	34.3	31.3	ASFV
			5 dpi	serum	25.4	−	24.3	32.6	ASFV
			6 dpi	serum	23.0	−	16.3	23.4	ASFV
			7 dpi	serum	17.2	−	15.7	23.7	ASFV
Pig	1806-7	Armenia/07	0 dpi	serum	−	−	−	34.7	negative
			1 dpi	serum	−	−	−	33.5	negative
			2 dpi	serum	39.0	−	−	34.3	negative
			3 dpi	serum	31.4	−	29.8	34.2	ASFV
			4 dpi	serum	18.9	−	20.8	28.8	ASFV
			5 dpi	serum	20.6	−	16.5	24.0	ASFV
			6 dpi	serum	20.8	−	15.9	24.0	ASFV
			7 dpi	serum	19.5	−	15.4	24.2	ASFV
WB	2002-1	Armenia/07	0 dpi	serum	−	−	−	34.8	negative
			2 dpi	serum	23.7	−	28.1	32.4	ASFV
WB	2002-2	Armenia/07	0 dpi	serum	−	−	−	35.6	negative
			2 dpi	serum	31.2	−	35.9	35.1	ASFV
			4 dpi	serum	17.9	−	21.0	24.3	ASFV
WB	2002-3	Armenia/07	0 dpc	serum	−	−	−	34.4	negative
			7 dpc	serum	31.2	−	38.0	35.0	ASFV
			9 dpc	serum	17.8	−	20.1	24.0	ASFV
			13 dpc	serum	20.8	−	22.3	27.9	ASFV
WB	2002-4	Armenia/07	0 dpc	serum	−	−	−	36.3	negative
			7 dpc	serum	−	−	−	35.4	negative
			9 dpc	serum	−	−	−	34.1	negative
			11 dpc	serum	−	−	−	34.2	negative
			14 dpc	serum	21.9	−	26.3	30.4	ASFV
WB	2002-5	Armenia/07	0 dpc	serum	−	−	−	35.6	negative
			7 dpc	serum	−	−	−	34.9	negative
			9 dpc	serum	25.6	−	28.9	32.6	ASFV
Pig	1705-1	Armenia/07	5 dpi	spleen	23.1	−	23.4	25.7	ASFV
Pig	1705-2	Armenia/07	6 dpi	spleen	23.3	−	23.7	25.2	ASFV
Pig	1705-3	Kenya/05	6 dpi	spleen	24.0	−	23.9	25.2	ASFV
Pig	1705-4	Kenya/05	6 dpi	spleen	24.2	−	24.8	24.3	ASFV
Pig	1705-5	Espana/75	6 dpi	spleen	22.6	−	22.0	24.4	ASFV
Pig	1705-6	Espana/75	5 dpi	spleen	23.8	−	21.7	24.1	ASFV
Pig	2105-9	AQS-C-1-22	8 dpi	tonsil	27.7	−	29.5	25.9	ASFV
				spleen	21.6	−	24.9	24.8	ASFV
				kidney	27.4	−	27.6	19.1	ASFV
				mesenteric LN	27.2	−	29.0	22.4	ASFV
Pig	2105-10	AQS-C-1-22	9 dpi	tonsil	16.9	−	22.7	24.0	ASFV
				spleen	19.5	−	23.0	25.5	ASFV
				kidney	22.9	−	25.9	23.4	ASFV
				mesenteric LN	20.9	−	25.8	25.1	ASFV
WB	2101-1	Armenia/07	9 dpi	tonsil	21.2	−	22.1	25.6	ASFV
				spleen	21.4	−	22.3	27.5	ASFV
				kidney	25.8	−	25.5	28.0	ASFV
WB	2101-2	Armenia/07	13 dpi	tonsil	20.5	−	22.9	26.6	ASFV
				spleen	17.0	−	20.4	26.5	ASFV
				kidney	24.3	−	24.8	26.4	ASFV
WB	2101-3	Armenia/07	7 dpi	tonsil	21.4	−	21.8	24.4	ASFV
				spleen	19.9	−	20.8	26.4	ASFV
				kidney	22.2	−	24.6	25.9	ASFV

dpi, days post-inoculation; dpc, days post-contact; LN, lymph node; –, not detected; WB, wild boar. * Ct values were indicated.

**Table 2 viruses-14-00498-t002:** Detection of CSFV in sera obtained from pigs and pig-boar hybrids experimentally infected with CSFV Japanese isolates.

					Multiplex rRT-PCR ^b^			
		Inoculated	Date of Sampling	RT-PCR ^a^ (5′-UTR)	CSF	ASF	IC	
Animal	ID	Virus Strain			Cy5	FAM	VIC	Decision
Pig	1810-1	JPN/1/2018	0 dpi	−	−	−	35.1	negative
			1 dpi	−	−	−	34.8	negative
			2 dpi	−	−	−	34.7	negative
			3 dpi	−	−	−	35.9	negative
			4 dpi	+	36.9	−	35.8	CSFV
			5 dpi	+	32.4	−	35.3	CSFV
			6 dpi	+	29.3	−	35.1	CSFV
			7 dpi	+	26.3	−	35.1	CSFV
Pig	1810-2	JPN/1/2018	0 dpi	−	−	−	34.9	negative
			1 dpi	−	−	−	35.8	negative
			2 dpi	−	−	−	33.4	negative
			3 dpi	−	36.8	−	36.0	CSFV
			4 dpi	+	34.9	−	35.8	CSFV
			5 dpi	+	32.3	−	35.1	CSFV
			6 dpi	+	29.6	−	35.1	CSFV
			7 dpi	+	27.3	−	35.0	CSFV
pig-boar hybrid	1905-4	JPN/27/2019	0 dpi	−	−	−	35.6	negative
			1 dpi	−	−	−	35.7	negative
			2 dpi	−	−	−	36.1	negative
			3 dpi	−	−	−	34.1	negative
			4 dpi	−	33.9	−	38.3	CSFV
			5 dpi	+	31.7	−	35.6	CSFV
			6 dpi	+	29.0	−	34.5	CSFV
			7 dpi	+	26.7	−	36.0	CSFV
			8 dpi	+	24.2	−	37.2	CSFV
			10 dpi	+	22.9	−	34.7	CSFV
			14 dpi	+	21.1	−	37.3	CSFV
			28 dpi	+	27.8	−	37.2	CSFV
pig-boar hybrid	1905-5	JPN/27/2019	0 dpi	−	−	−	33.9	negative
			1 dpi	−	−	−	35.6	negative
			2 dpi	−	−	−	35.3	negative
			3 dpi	−	37.4	−	36.0	CSFV
			4 dpi	−	35.2	−	35.8	CSFV
			5 dpi	+	32.6	−	34.3	CSFV
			6 dpi	+	29.2	−	36.3	CSFV
			7 dpi	+	26.2	−	37.5	CSFV
			14 dpi	+	24.7	−	36.9	CSFV

dpi, days post-inoculation; −, not detected. ^a^ Results of RT-PCR assay in this table are published [22,23]. ^b^ Ct values were indicated.

**Table 3 viruses-14-00498-t003:** Validation study for clinical samples from wild boars of the simple and multiplex real-time RT-PCR system for ASFV and CSFV.

		Crude NA Preparation & Multiplex rRT-PCR							
Serum		ASFV Positive		ASFV Negative					
		WB	Pig	WB	Pig	Sensitivity		Specificity	
Viral NA extraction & ASFV rPCR	Positive	8	9	0	1	94.4%	(17/18)	97.1%	(34/35)
	Negative	0	1	29	5				
		CSFV Positive		CSFV Negative					
		WB	Pig	WB	Pig	Sensitivity		Specificity	
Viral NA extraction & *pestivirus* RT-PCR	Positive	36	31	4	0	94.4%	(67/71)	91.9%	(34/37)
	Negative	2	1	27	7				
		Crude NA preparation & multiplex rRT-PCR							
Tissue homogenate		ASFV Positive		ASFV Negative					
		WB	Pig	WB	Pig	Sensitivity		Specificity	
Viral NA extraction & ASFV rPCR	Positive	9	14	0	0	100%	(23/23)	100%	(12/12)
	Negative	0	0	0	12				
		CSFV Positive		CSFV Negative					
		WB	Pig	WB	Pig	Sensitivity		Specificity	
Viral NA extraction & *pestivirus* RT-PCR	Positive	14	10	0	0	100%	(24/24)	100%	(12/12)
	Negative	0	0	0	12				

NA, nucleic acid; WB, wild boars and pig-boar hybrids.

## Data Availability

Not applicable.

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
