# Peer review of "Establishment of a Direct PCR Assay for Simultaneous Differential Diagnosis of African Swine Fever and Classical Swine Fever Using Crude Tissue Samples"

_viruses, 2022, doi:10.3390/v14030498_

Round 1
Reviewer 1 Report
The article identifies a lysis buffer that can function with a single step rRT-PCR without further purification for simultaneous detection of ASFV and CSFV in organ tissue homogenates and serum. Preparation takes 5 minutes and the PCR itself is 55 minutes. This assay can be another important tool for diagnosis and differentiation of ASFV and CSF from clinical samples. When serum was used, the assay showed 100% and 98.6% sensitivity and 97.9% and 91.9% specificity for ASFV and CSFV detection. (Please check the calculations again as you have false negatives with the serum samples)
Overall, this is an interesting approach, though the concept itself is not novel as many publications exist establishing a PCR after introducing samples to a lysis buffer for other pathogens. It is interesting to see that this worked with tissue homogenates. Have you looked at the stability of the sample once it is left in the lysis buffer? Can the samples be put at 4C or -20C once in this lysis buffer for repeated testing?
Authors state their assay is novel, however, two recent publications on ASFV utilizing direct PCR methods with samples other than organs that come to mind are:
Elnagar, A.; Pikalo, J.; Beer, M.; Blome, S.; Hoffmann, B. Swift and Reliable “Easy Lab” Methods for the Sensitive Molecular Detection of African Swine Fever Virus. Int. J. Mol. Sci. 2021, 22, 2307. https://doi.org/10.3390/ijms22052307
Elnagar, A.; Harder, T.C.; Blome, S.; Beer, M.; Hoffmann, B. Optimizing Release of Nucleic Acids of African Swine Fever Virus and Influenza A Virus from FTA Cards. Int. J. Mol. Sci. 2021, 22, 12915. https://doi.org/10.3390/ijms222312915
Suggestion: use the term direct PCR for PCR of crude samples to improve searches for your publication. For example “Establishment of a Direct PCR Assay for simultaneous ….”
Line 98 How did you clear the supernatant? Centrifugal force (rcf), time?
Line 122 Did you bleed from the Jugular vein?
Line 124 How were the organs emulsified and with what device, centrifugal force, and length of time?
Line 316 Calculation of sensitivity for serum samples. You had one positive pig test negative for ASFV. So how can you have 100% sensitivity? The same for CSFV with 3 animals falsely testing negative. Please check the calculations again
For ASFV Serum: Sensitivity TP/(TP+FP) =17/(17+1) 17/18=94.44%
For CSFV Serum: Sensitivity TP/(TP+FP) = 67 /70=95.7%
For CSFV Serum: Specificity TN /(TN+TP) 34/38=89.4%
Line 354 The authors state their methods are a remarkable improvement without evidence of comparing data to other standard ASFV CSFV diagnostic PCR assays. This is an overstatement without any direct comparison.
Author Response
Dear Reviewer 1,
We are grateful to your useful comments and suggestions, which have helped us improve our manuscript considerably. We have revised the manuscript based on the comments raised. We have also provided our point-by-point responses to each of the comments below:
Comment 1) Have you looked at the stability of the sample once it is left in the lysis buffer? Can the samples be put at 4C or -20C once in this lysis buffer for repeated testing?
Answer) Thank you for your interest. The storage stability of the crude nucleic acid sample depends on the RNase content and the condition of the sample itself. Therefore, the authors recommend that storage after mixing with the lysis buffer should be avoided and subjected to the PCR reaction immediately.
Comment 2) Authors state their assay is novel, however, two recent publications on ASFV utilizing direct PCR methods with samples other than organs that come to mind are:
Elnagar, A.; Pikalo, J.; Beer, M.; Blome, S.; Hoffmann, B. Swift and Reliable “Easy Lab” Methods for the Sensitive Molecular Detection of African Swine Fever Virus. Int. J. Mol. Sci. 2021, 22, 2307. https://doi.org/10.3390/ijms22052307
Elnagar, A.; Harder, T.C.; Blome, S.; Beer, M.; Hoffmann, B. Optimizing Release of Nucleic Acids of African Swine Fever Virus and Influenza A Virus from FTA Cards. Int. J. Mol. Sci. 2021, 22, 12915. https://doi.org/10.3390/ijms222312915
Suggestion: use the term direct PCR for PCR of crude samples to improve searches for your publication. For example, “Establishment of a Direct PCR Assay for simultaneous ….”
Answer) Thank you for your suggestion. According to your comment, the title and abstract were revised (line 2, 18 in the revised manuscript).
Comment 3) Line 98 How did you clear the supernatant? Centrifugal force (rcf), time?
Answer) The tissue samples were centrifuged at 8,000×g for 10 min and their cleared supernatants were aliquoted. This information was added in the revised manuscript (Lines 99-100).
Comment 4) Line 122 Did you bleed from the Jugular vein?
Answer) According to your suggestion, the word was changed to Jugular vein in the revised manuscript (Line 122).
Comment 5) Line 124 How were the organs emulsified and with what device, centrifugal force, and length of time?
 Answer) The tissues were homogenized using a Micro Smash MS-100R (TOMY SEIKO, Japan) and emulsified in Dulbecco’s modified Eagle’s medium to yield a 10% (wt/vol) tissue homogenate. The homogenates were cleared by centrifugation at 8, 000×g for 10 min at 4oC. This information was added in the revised manuscript (Lines 124-127).
Comment 6) Line 316 Calculation of sensitivity for serum samples. You had one positive pig test negative for ASFV. So how can you have 100% sensitivity? The same for CSFV with 3 animals falsely testing negative. Please check the calculations again
For ASFV Serum: Sensitivity TP/(TP+FP) =17/(17+1)=17/18=94.44%
For CSFV Serum: Sensitivity TP/(TP+FP) =67/70=95.7%
For CSFV Serum: Specificity TN/(TN+TP) =34/38=89.4%
Answer) Thank you for your comment. The sensitivities were re-calculated by the following and rewritten in the revised manuscript (Line 23, 324, Table 3).
For ASFV Serum: Sensitivity TP/(TP+FN) =17/(17+1)=17/18=94.4%
For CSFV Serum: Sensitivity TP/(TP+FN) =67/(67+4)=67/71=94.4%
For CSFV Serum: Specificity TN/(TN+FP) =34/(34+3)=34/37=91.9%
Comment 7) Line 354 The authors state their methods are a remarkable improvement without evidence of comparing data to other standard ASFV CSFV diagnostic PCR assays. This is an overstatement without any direct comparison.
Answer) The method which we established in the present study provides a simple procedure for nucleic acid extraction within only 5 min and enables simultaneous diagnosis of ASF and CSF caused by DNA and RNA viruses, respectively, with considerable reliability. In addition, this method requires only a small amount of sample and employs techniques to minimize both false-negative and positive results as described in the manuscript (Lines 321-336). We believe this method significantly enhance the rapid diagnosis and field surveillance of both diseases in endemic areas such as Japan and other Asian countries. Considering the above and your comment, the sentence was revised in the revised manuscript (Line 356).
Reviewer 2 Report
African Swine Fever (ASF) and Classical Swine Fever are the biggest concerning in the pork industry, causing economical loss and raising frights on international trade of pork products. Clinically ASF and CSF are indistinguible, therefore is imperative an easy and practical diagnosis for both diseases. This article raises important information about a novel method to differentiate ASF from CSF. The data are relevant and elegantly written, however there are some minor gaps that need to be addressed.
Minor:
- Line 87-88: Please indicate the source of CPK cell culture
- Line 133: Please format “25 oC”
- Please include statistical analysis methods.
- what test was applied to check the correlation among the tests?
-what test the authors take to choose the sensitivity and specificity?
- Line 324: Extra space in the sentence: “For example”
Author Response
Dear Reviewer 2,
We are grateful to your useful comments and suggestions, which have helped us improve our manuscript considerably. We have revised the manuscript based on the comments raised. We have also provided our point-by-point responses to each of the comments below:
Comment 1) Line 87-88: Please indicate the source of CPK cell culture
Answer) According to your comment, the source was indicated in the revised manuscript (Line 87, Reference No. 19).
Comment 2) Line 133: Please format “25 oC”
Answer) According to your comment, it was revised in the revised manuscript (Line 135).
Comment 3) Please include statistical analysis methods.
- what test was applied to check the correlation among the tests?
-what test the authors take to choose the sensitivity and specificity?
Answer) The rate of correlation coefficient and efficiency of amplification were calculated using 7500 Software v2.3 (Life Technologies). The statistical sensitivity and specificity were calculated as follows; Sensitivity = TP/(TP + FN), Specificity = TN/(TN + FP). (True positive (TP), a positive result in both methods; False positive (FP), a positive result in the multiplex rRT-PCR but a negative result in primary diagnostic method; True negative (TN), a negative result in both methods; False negative (FN), a negative result in the multiplex rRT-PCR but a positive result in primary diagnostic method). This information was indicated in the revised manuscript (Lines 193-194, 312-318).
Comment 4) Line 324: Extra space in the sentence: “For example”
Answer) According to your comment, it was revised in the revised manuscript (Line 333).
Reviewer 3 Report
Introduction, line 66: Change "respectively, also naturally" to "respectively, but can also naturally".
Figure 2: I suggest having only the bullets no lines, since it is different samples.
Figure 4: You never indicate the multiplex results graphs. As shown in this study all results are simplex PCRs can you include multiplex graphs as indicated with controls
Table 2: cannot find your note b link in table
Table 3: You have one serum for ASF neg pig from positive group, which makes sensitivity 17/18 and 94.44%. You also have for CSF serum 67/71, which gives sensitivity of 95.71%. Please check CSF serum specificity as well.
Discussion, line 321: "taxing" might not be correct word in context use.
Discussion, line 344: "false negative results for CSFV", is true but what about your one ASFV positive pig?
Author Response
Dear Reviewer 3,
We are grateful to your useful comments and suggestions, which have helped us improve our manuscript considerably. We have revised the manuscript based on the comments raised. We have also provided our point-by-point responses to each of the comments below:
Comment 1) Introduction, line 66: Change "respectively, also naturally" to "respectively, but can also naturally".
Answer) According to your comment, the sentence was revised in the revised manuscript (Line 65).
Comment 2) Figure 2: I suggest having only the bullets no lines, since it is different samples.
Answer) According to your comment, Figure 2 was revised to show only the bullets with no lines.
Comment 3) Figure 4: You never indicate the multiplex results graphs. As shown in this study all results are simplex PCRs can you include multiplex graphs as indicated with controls
Answer) The representative multiplex results graphs of the established method were indicated in Figure 4.
Comment 4) Table 2: cannot find your note b link in table
Answer) The b link is indicated with “multiplex rRT-PCR” located at the top of the Table 2. Could you check under the line 295 in the revised manuscript?
Comment 5) Table 3: You have one serum for ASF neg pig from positive group, which makes sensitivity 17/18 and 94.44%. You also have for CSF serum 67/71, which gives sensitivity of 95.71%. Please check CSF serum specificity as well.
Answer) Thank you for your comment. The sensitivities were re-calculated and rewritten in the revised manuscript (Line 23, 324, and Table 3). In addition, the calculation method of sensitivity and specificity were clearly indicated in the revised manuscript (Lines 312-318).
Comment 6) Discussion, line 321: "taxing" might not be correct word in context use.
Answer) According to your comment, the sentence was revised in the revised manuscript (Line 330).
Comment 7) Discussion, line 344: "false negative results for CSFV", is true but what about your one ASFV positive pig?
Answer) As described in Result section, there was 4-fold difference in the input amount between crude and purified nucleic acid preparations in proportion to the volume of the serum used for the purification. And the Ct value for ASF of the serum of pig ID 1806-7 at 2 dpi (which was ASF negative in the multiplex rRT-PCR) in the singleplex ASFV rPCR was 39.0 indicating very few copies of viral gene contained. This should be the reason for generating one false-negative result for ASFV. This explanation was added in the sections of Result and Discussion in the revised manuscript (Lines 256-258, 354-355).